# Calibrating low-cost rain gauge sensors for their applications in IoT infrastructures to densify environmental monitoring networks

Robert Krüger[1], Pierre Karrasch[2], Anette Eltner[1]

[1]Juniorprofessur für Geosensorsysteme, TU Dresden, Dresden, 01062, Germany
[2] Sächsisches Landesamt für Umwelt Landwirtschaft und Geologie, Dresden, 01326, Germany

*Correspondence to*: Robert Krüger (robert.krueger@tu-dresden.de)

**Abstract.** Environmental observations are crucial for understanding the state of the environment. However, current observation networks are limited in spatial and temporal resolution due to high costs. For many applications, data acquisition with a higher resolution would be desirable. Recently, Internet of Things (IoT) -enabled low-cost sensor systems offer a
solution to this problem. While low-cost sensors may have lower quality than sensors in official measuring networks, they can still provide valuable data. This study describes the requirements for such a low-cost sensor system, presents two implementations, and evaluates the quality of the factory calibration for a widely used low-cost precipitation sensor. Here, twenty sensors have been tested for an 8-month period against three reference instruments at the meteorological site of the TU Dresden. Further, the factory calibration of 66 rain gauges has been evaluated in the lab. Results show that the used sensor
falls short for the desired out-of-the-box use case. Nevertheless, it could be shown that the accuracy could be improved by further calibration.

## 1    Introduction

Environmental observations are a pillar of environmental science. They provide the necessary data to describe and model the state of the environment and its spatial and temporal changes. Furthermore, the data collected can be used to identify and assess
possible natural risks and thus warn of potential natural hazards. Environmental observations also form the basis for decision-making in environmental policy and for monitoring the outcome of the resulting measures, which requires reliable and systematically collected data. In line with this requirement, data on climate, soil, water balance and air quality are collected in many countries by authorities that operate permanent monitoring networks (Kaspar et al., 2013). The stations within these monitoring networks are usually equipped with professional measuring tools, which, like the sites themselves, meet certain
standards of the respective international organisations. Furthermore, the operation and maintenance of such monitoring networks is ensured by a high level of human resources. And the measured values are subjected to quality control. The resulting costs lead to observation networks that cannot be condensed indefinitely, even if observations in higher spatial and temporal resolution would be desirable for many applications, e.g., such as warning of flash floods or landslides (Lobligeois et al., 2014; Gamperl et al., 2021).

Developments over the last two decades in the field of the Internet of Things (IoT) allow this shortcoming of institutional measurement networks to be addressed. The availability of ever smaller, cheaper, more power-efficient devices and sensors combined with the ubiquitous availability of connectivity to the internet make it possible to collect and process data where it is needed. Even if the quality and reliability of such devices is lower than that of official measuring stations, the resulting data sets with higher spatial and temporal resolution can represent added value. For instance, the use of IoT-enabled sensors allows

further automation and fine-tuning of agricultural processes by collecting data on climate (e.g. $CO_2$ emissions (Brown et al., 2020)) or soil (e.g. moisture (Adla et al., 2020)). In the field of natural hazard research low-cost GNSS and accelerometer devices communicating within an IoT network are used to monitor boulders (Dini et al., 2021). Another study illustrates the suitability of IoT devices measuring water level and soil moisture to monitor floods during tropical storms (Mendoza-Cano et al., 2021).

In regard of precipitation monitoring, low-cost weather stations (PWS - personal weather stations) fill gaps in measurement networks that were previously interpolated or determined by radar and satellite products (with corresponding inaccuracies) (Fraga et al., 2019; de Vos et al., 2017). Although many studies have shown that the use of IoT-enabled low-cost precipitation sensors is possible (Lopez and Villaruz, 2015; Rodríguez et al., 2021), it is necessary to ensure that the data generated meets the necessary requirements for data quality when used in addition to official networks. This can be achieved through the

calibration of low-cost sensors - however, this step is time-consuming (Humphrey et al., 1997) and hence costly. Therefore, calibration runs are counter-productive to the intended purpose of low-cost sensor technology. This provokes the question if it is suitable, to actually use low-cost sensors relying only on the factory calibration of the manufacturer. To arrive at a conclusion about the quality of factory calibration, it is thus necessary to evaluate a larger number of sensors of the same type. In many studies utilizing low-cost sensors, only a single or a very small number of instruments have been used, not yet addressing this

question (Fraga et al., 2019; Strigaro et al., 2019; Sudantha et al., 2019; de Vos et al., 2017).

In this study, we describe the requirements for a low-cost sensor system and show two implementations using open hardware. Furthermore, we analyse the quality of factory calibration for a widely used low-cost precipitation sensor and thus test its suitability for an out-of-the-box use. Measurement campaigns were conducted both in the laboratory and in the field.

## 2    Methods

### 2.1    Requirements for low-cost sensor systems

To improve the resolution of any official environmental measurement network, the sensor systems have to fulfil different requirements. When using a high number of sensor systems, they have to be **low-cost** while maintaining a certain level of **data quality** and **reliability** to ensure an effective application. Thus, sensors have to be quality checked before being used. To further reduce costs, the sensor system should be robust and low-maintenance.

The proposed systems should be **energy efficient** so that the systems can operate for long periods of time without replacement or can be charged by solar panels. This would make the systems independent from being connected to the power grid, which maximises the possibilities for measuring sites. For real- or near real-time use of the data, the use of **wireless connectivity** is required to transmit the data from the sensors to the users. This also improves flexibility in the selection of measuring sites. The sensor system should be **easy to** install, **use** and maintain, ideally even by people not familiar with the subject. This also

enables the use of volunteers (citizen scientists) to further reduce costs. Since not all requirements have to be met at all locations, **modularity** of the system would be desirable; be it the choice of sensors, power supply or connectivity. Further, to make the system as applicable and transferable as possible, **open-source hardware** should be used.

## 2.2    Development of a low-cost sensor system for data acquisition

When designing a low-cost sensor system, one can either use hardware specifically designed for the use case, or rely on widely

available open-source hardware. The latter has the advantage to make the system as applicable and transferable as possible. Two widely used open-source options are either the Raspberry Pi (RPi) or the Arduino ecosystem.

The Raspberry Pi is a single board computer, that is widely used for a variety of applications such as education, home media centres, home automation or IoT projects. The boards can be connected to a wide range of sensors, actuators, and other electronic components through 28 digital input/outputs pins using standard connectors and protocols (SPI, I2C, Serial).

Furthermore, cameras can also be connected to the RPi and therefore used for environmental monitoring, e.g., to measure water levels (Eltner et al., 2018) or to detect rockfalls (Blanch et al., 2020). On the RPi, Raspbian – a Linux variant is used as operating system, while the connected sensors can be controlled and read out e.g., using Python. Since the RPis are real computers, the sensor data or captured images can also be processed directly on them. Most RPi Boards provide connectivity through an integrated wifi/bluetooth module, but other types of connectivity can be established either via USB (e.g., UMTS

Dongle) or specific shields connected to the I/O Pins.

Arduino is an open-source electronics platform based on easy-to-use hardware and software. Unlike the Raspberry Pi, the Arduino Ecosystem consists of boards featuring different microcontrollers and are not fully-fledged computers. Nevertheless, they provide almost the same connectivity to sensors via the same connectors and protocols as used with the Raspberry Pi. While the computing abilities on the board are limited compared to the RPi, their energy efficiency is significantly higher.

Arduino boards with different options of integrated network connectivity (e.g. LoRa, GSM, Narrowband IoT (SigFox)) are available (Singh et al., 2020).

Overall, the choice between the Raspberry Pi and Arduino will depend on the specific requirements of the use case and the conditions available at the measuring site. If on-site processing of images is needed or power supply is available, the Raspberry Pi is a good choice. If one needs to read and broadcast sensor data from remote locations an Arduino is a more suitable, hence

a more energy efficient and cost-effective option. Thus, two different modular sensor systems are proposed, which are based on the two different platforms.

### 2.2.1 Raspberry Pi

The Raspberry Pi Zero W (RPi0W) was chosen to keep energy consumption and acquisition costs low. In previous studies, it has been proven suitable to measure hydrological parameters, e.g., via a camera gauge (Eltner et al., 2021). The RPi0W model combines the lowest price (sub 10€) with the lowest energy consumption (0.75-1.5W) within the available Raspberry Pi models, while still being fast enough for e.g. image acquisition.

The RPi based model is powered through the mains, which allows for running it constantly, as energy consumption is not a critical factor. Thus, logging of sensor data in a high resolution and uploading it in a sufficient schedule is possible. While it is possible to power the Raspberry Pi system using a solar panel and battery, the energy consumption of 0.5 to 5 Watts (depending on the performed processes and chosen model) may require a large panel and battery generating higher costs.

On the RPi, the sensors were connected to the I/O Pins and read out via small python scripts. For many sensors, libraries for Python are available, making communication between sensor and RPi easy to set up. Read out data was then written into a SQLite database file. Depending on the use case, data can be instantly transmitted or read out periodically from the database and subsequently uploaded into a cloud infrastructure. Here, the Message Queuing Telemetry Transport (MQTT), a lightweight message transport protocol, was used. Data transfer was realized either through the on-board wifi module, or using a USB-UMTS modem utilizing a dial-up internet connection. This connection is also used for setting and frequently updating the system time of the RPi via the internet, as the RPI has no built-in real-time clock (RTC).

### 2.2.2 Arduino

The proposed Arduino system consists of an Arduino Board from the Arduino MKR series, which uses a low power ARM Cortex-M0 SAMD21 processor. Specifically, the Arduino MKR Fox 1200 and the Arduino MKR GSM 1400 were used – other network options are available. The main benefit of the proposed system is the reduced energy consumption compared to the RPi system. As the energy consumption in the active state (measuring or transmitting sensor data) is already 80% lower compared to the RPi, the energy consumption of the whole system can be reduced to less than 5 mW by utilizing deep sleep modes through a low power-library. The Arduino system can be run directly from two 1.5 V batteries or a small solar panel. We achieved running times of up to six months on a pair of D-Cells and basically unlimited running time on a small solar panel and a LiPo Cell for buffering.

The Arduino system requires two additional parts – namely a standard SD card for data storage, and an external real-time clock (RTC) module. For the MKR Fox 1200 version of the Arduino system this creates a limitation of the system as the time cannot be updated online through the SigFox network. Thus, the accuracy of the timestamps relies on the drift of the RTC module. Here the DS3231 was used, which can have a drift of up to 2ppm (maxim integrated, 2015), which equals to about 63 seconds over the course of a year.

Another limitation of the MKR Fox 1200 version is the bandwidth and message limit the used ISM-band possess (this also applies to e.g., LORA-Wan). Here, only 140 messages of 12 bytes a day are allowed for transmission, which might be a

bottleneck if several sensors are used. So, if data (and transfer) with a high temporal resolution is needed, one has to consider

a GSM based solution.

### 2.2.3    System cost

In this study, the costs for the cheapest monitoring solution is the described RPi option, powered through the mains while transferring the measured data via WLAN (about 40 Euro + sensors). The cheapest off-grid solution (system based on Arduino MKR Fox 1200) costs about 75 Euros + sensors. A detailed summary of system configurations and corresponding costs is

given in Table 1.

**Table 1. Cost for different sensor systems depending on use case**

| | | | solar powered | battery powered | solar powered |
|---|---|---|---|---|---|
| **electricity** | x | x | | | |
| **wifi** | x | | | | (x) + camera |
| **system** | RPi ZeroW 10 euro | RPi ZeroW 10 euro | Arduino MKR Fox 45 euro | Arduino MKR Fox 45 euro | RPi ZeroW 10 euro |
| **power supply** | power adapter 10 euro | power adapter 10 euro | 5W solar panel + battery loading circuit 30 euro | D-Cells 5 euro | 50W+ solar panel + battery loading circuit 100+ euro |
| **accessories** | SD card housing 20 euro | SD card GSM modem housing 35 euro | SD card RTC module housing 25 euro | SD card RTC module housing 25 euro | SD card housing (GSM modem) 20/35 euro |
| **cost total** | 40 euro | 55 euro | 100 euro | 75 euro | 130+/145+ euro |

### 2.2.4 Low-cost environmental sensors

The developed systems are capable of connecting to widespread types of low-cost sensors (e.g., the temperature/humidity sensors *DHT22* and *SHT31*). In this study, a *Bosch BME280* weather sensor (i.e., temperature, humidity, air-pressure) and the semi-professional *Davis Vantage Pro2* rain gauge had been used for demonstration. Further, a Raspberry Pi based system has been used in conjunction with a SHT31 temperature/humidity sensor, a pyranometer and a Davis Vantage Pro2 rain gauge. The source code for those setups is available through Zenodo (Krüger, 2024).


### 2.2.5 Low–cost rain gauge

The considered rain gauge is using the tipping bucket principle. It consists of a collector cone with a spherical opening on the top. The collected precipitation is lead through a debris-filtering screen into a container with two buckets on a pivot. When one bucket fills up with water, it eventually tips and empties the container, bringing the other bucket in position to be filled.

Each time the bucket tips, a contact is triggered, which can be counted by a connected microcontroller or sensor system. Within the timeframe of this study, the design of the *Davis* rain gauge has been altered by the manufacturer. In particular, three different models were available in that timeframe. In the first update the design of the upper part (collector cone) of the device was changed. While the old version (model nr. 7852M/7857M) had the shape of a truncated cone, the new one (model nr. 6463M/6465M further referenced as TYPE A) has the shape of the funnel inside the cone. After the field study described in

the next chapter, the design (model nr. 6464M/6466M further referenced as TYPE B) was changed again. This time the tipping mechanism was changed from a two-bucket design into a one-bucket design. Instead of tipping and bringing the second bucket into position, the new design uses a counterweight, that returns the bucket back into the collection position after emptying when reaching the given amount of precipitation. The three types can be seen in **Fehler! Verweisquelle konnte nicht gefunden werden.**. As the older types are still widely in use, the results of this study still apply for a widespread of users.

## 2.3 quality assessment of a low-cost rain gauge

To draw a benefit from the use of low-cost sensors, these have to provide a certain level of data quality and reliability. Thus, these properties have to be assessed and verified for a given sensor type. Two goals were pursued in regard of the performance assessment: On the one hand, the accuracy of the rain gauge was assessed. On the other hand, the quality (spread) of the factory calibration and thus the suitability in a low-cost, out-of-the-box use case was examined. Therefore, two studies have been

performed, which will be described in the following sections.

### 2.3.1 Lab-calibration

To assess the quality of the factory calibration, a static calibration was carried out (Marsalek, 1981). Hereby the volume of water, which is required to tip the bucket, is measured using a syringe and a micro scale.

The calibration was carried out in the lab for 37 new tipping gauges of type TYPE A. Furthermore, 20 rain gauges of the same type used in the field study, were also examined directly at site, i.e., lab-calibrated onsite, after carrying out the field study. As the manufacturer changed the design after the completion of the field study, another lab-calibration with nine fabric new gauges of the type TYPE B was carried out.

**Table 2: Different versions of the used low-cost rain gauge: collector cone and measuring mechanism changed over time.**

| model | 7852M/7857M | 6463M/6465M (TYPE A) | 6464M/6466M (TYPE B) |
|---|---|---|---|
| collector cone | | | |
| measuring mechanism | | | |

In preparation of the lab-calibration the table was levelled utilizing adjusting screws at the table legs. The rain gauges themselves have been levelled using a built-in bubble level. Further, slices of paper have been used to account for remaining unevenness on the table. During the calibration, all gauges have been oriented in the exact same direction on the table.

All measurements were taken using a G&G micro scale with an accuracy of 0.01 g. The rain collector has a diameter of 16.5 cm, which equals to an amount of 4.277 ml or 4.269 g of water (Tanaka et al., 2001) for each tipping of the bucket, i.e., after 0.2 mm rain water has been collected, as stated by the manufacturer. For each gauge the following process was executed 20 times:

1. Calibration of the micro scale using a 100 g calibration weight.
2. Weighing of an arbitrarily chosen amount of water (4 - 7 g) on the micro scale.
3. Drawing up water into a syringe directly from the scale.
4. Slowly dripping water into one chamber of the tipping bucket (starting on the side of the pole mount) until the bucket tips.

5.        Clearing of the remaining water in the syringe back onto the micro scale and subsequently taking a weight measurement.

6.        Calculating the difference between the two measurements.

185        7.        Removing remaining water in the bucket using a paper towel.

8.        Back to Step 2.

Each weighing side was triggered ten times (for model 6463M/6465M) to minimise the influence of random measurement inaccuracies. In addition to calculating the weight differences, an iterated mean value of these differences was calculated. The deviation of this iterated mean value from the mean value after ten measurements provides information about the number of measurements after which the mean value calculated up to that point is stable, i.e., converged. This value can help to estimate how many measurements are necessary to make a reliable statement about the weighing properties of the precipitation tipping buckets. Further, the obtained mean allows statements on absolute accuracy to be made. According to the manufacturer, the expected value is 0.2 mm. Deviations from this value show whether over- or undercatch is to be expected in the precipitation measurement.

### 2.3.2     Field study

A field study was carried out at the meteorological site of the TU Dresden. The site is situated in the valley of the river Wilde Weißeritz (50°59' N, 13°35' E, altitude 220m NN, average annual precipitation in the period 1981/2010 was 795.4 mm (Tharandt Klimastation, 2023)). Several different professional instruments are measuring precipitation here, including a traditional Hellmann rain gauge, an *Ott Pluvio* gauge and a *Young* tipping gauge. These instruments are regularly maintained to meet the specifications of the World Meteorological Organization.

The Hellmann gauge is made of a steel cylinder and has a collecting area of 200 mm². The collected water runs through a funnel into a small container, which is emptied daily at 7 o'clock (CET) and the amount of water in the container measured to calculate the precipitation for the last 24 hours. The Hellmann device is used as the reference for the climatological measurements taken at this station since 1951 (Fig 1) and thus considered as the reference instrument in this study. The Hellmann gauge requires no mechanical and electronic parts, thus data quality should be stable, as the instrument has been set up properly.

The *Ott Pluvio* is a weighing rain gauge consisting of a collector cylinder (collecting area of 200 mm²) and weighing cell. The weight of the water collected in the cylinder is weighed constantly und subsequently a precipitation amount for each minute is generated. The resolution of the precipitation data is 0.1 mm.

The third professional device is a *Young* tipping gauge – utilizing the same measurement principle as the low-cost gauge, i.e., using tipping bucket. The *Young* gauge also has a collecting area of 200 mm² and it has a measurement resolution of 0.1 mm per tip. In contrast to the low-cost gauges, this device is equipped with a heating, which allows for snow precipitation to be measured.

Even with three professional gauges, the true precipitation remains unknown as each of these devices area also associated with uncertainties. Triple collocation analysis (Stoffelen and Vogelzang, 2012; Stoffelen, 1998) can be used to estimate the error variances of these three independent, but collocated datasets without requiring knowledge of the true precipitation amount. We used a longer time series of the three professional gauges lasting three years (from 01.01.2017 to 31.12.2019), consisting of daily observations, to estimate the uncertainties. Inspection of the three scatterplots (Figure 1) with all combinations of

reference gauges led to the assumption that the *Ott Pluvio* is the best performing, because the scatterplots for the *Hellmann and Young* devices revealed lower correlations (Stoffelen and Vogelzang, 2012). Therefore, we used an implementation provided by Jur Vogelzang[1] to estimate the error variances with the *Ott Pluvio* as reference system. Subsequently, the daily error standard deviations could be determined: stdPluvio = 0.150mm d$^{-1}$, stdHellmann = 0.183mm d$^{-1}$, stdYoung = 0.278mm d$^{-1}$. These values were used to evaluate the results of the low-cost gauges when compared with the reference gauges.

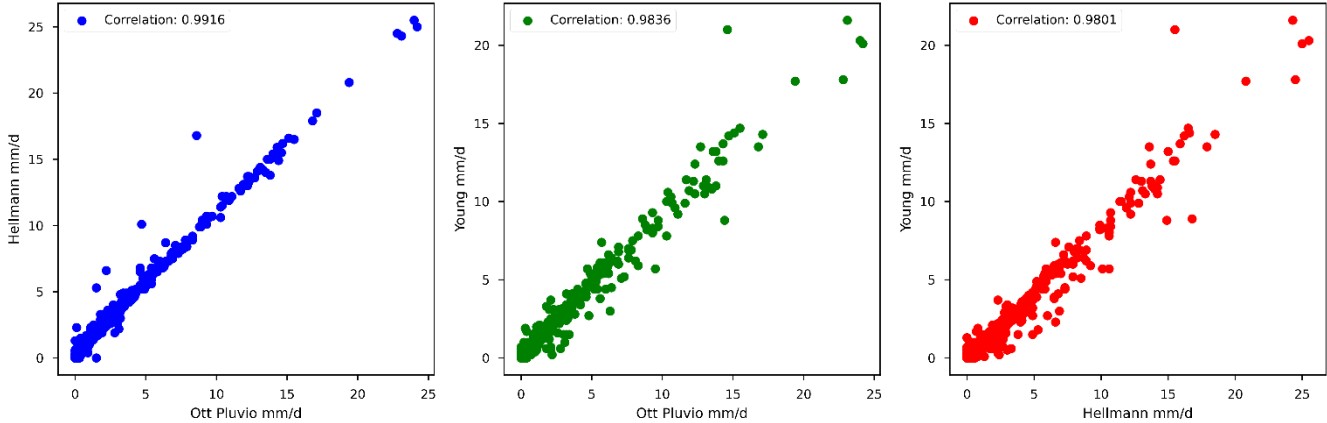


**Figure 1: Scatterplots for the combinations of the 3 professional gauges depicting daily precipitation values for 2017 to 2019**

The setup of the study was chosen to enable the analysis of the spread of measurements relying on the factory calibration. Therefore, an array of 20 identical, fabric new, low-cost rain gauges of the TYPE A were set up. The rain gauges were mounted

on a wooden frame in an array of four by five devices, covering an area of about 1 m². The frame is levelled and set at a height of about 1 m above ground to match to the height of the reference gauges, located about 10 m to the south of the reference instruments. Horizontality of the rain gauges was ensured by the usage of the built-in bubble level of the rain gauges and the washers while fixing the gauges to the (levelled) frame. The setup is shown in Figure 2. A Raspberry Pi (RPi Modell 3b) sensor system is used to log the precipitation data (i.e., timestamps of tipping events) of the rain gauges. To compare the low-cost

gauge measurements with the reference instruments, the amount of tipping events is multiplied by 0.2 mm per tipping, as stated by the manufacturer. Time synchronisation of the low-cost sensor system with the reference gauges was ensured by setting the

---

[1] https://github.com/knmiscat/triple_collocation

time for both type of systems through the network time protocol (NTP). Cumulative rainfall was measured in the months from August 2018 to April 2019.

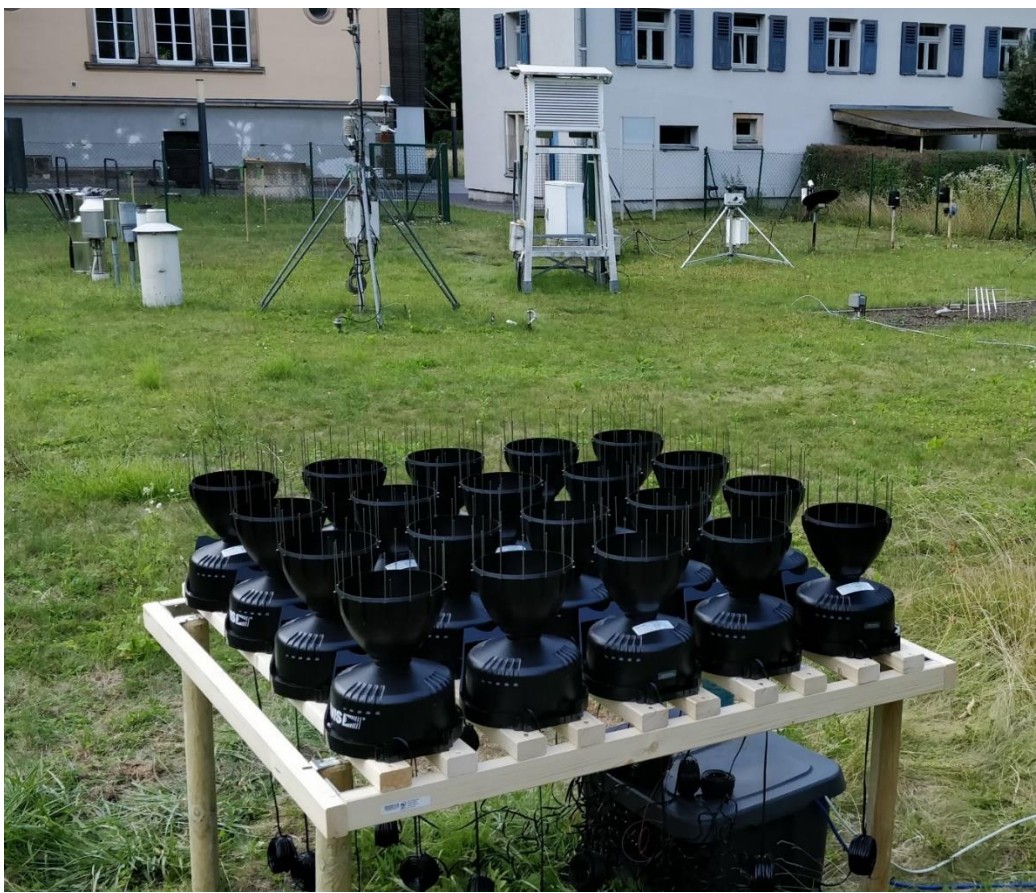


**Figure 2: Tharandt meteorological site with reference instruments (in background on the left side) and field study setup (in foreground)**

## 3    Results

### 3.1    Lab-calibration

In total, 66 rain gauges have been tested. Of these devices, 37 were fabric new gauges of TYPE A and 20 had been used already for about half a year (i.e., in the field study –TYPE A). Further nine gauges used the new single tipping bucket mechanism (TYPE B). None of the gauges had been recalibrated before (Krüger, 2024).

Considering all rain gauges of TYPE A, the mean water amount required for one tip was 0.174 mm with a standard deviation of 0.013 mm. The mean value for new gauges was 0.175 mm (std. dev. 0.014 mm) and amounted for the used gauges to a

mean of 0.172 mm (std. dev 0.009 mm) (Fig. 2). In contrast, the amount of water required for one tip for gauges of TYPE B

was 0.194 mm with a standard deviation of 0.004 mm. The distribution of results in all four groups (A – new, A – used, A – all, B) was tested if it is Gaussian, using the Kolmogorov–Smirnov test (alpha = 0.05). This could be confirmed for all four groups.

Confidence intervals for a level of 95% have been calculated as follows: A-new: 0.1680 mm - 0.1795 mm; A-used:
0.1699 mm - 0.1795 mm; A-all: 0.1704 mm - 0.1771 mm; B: 0.1911 mm - 0.1972 mm).

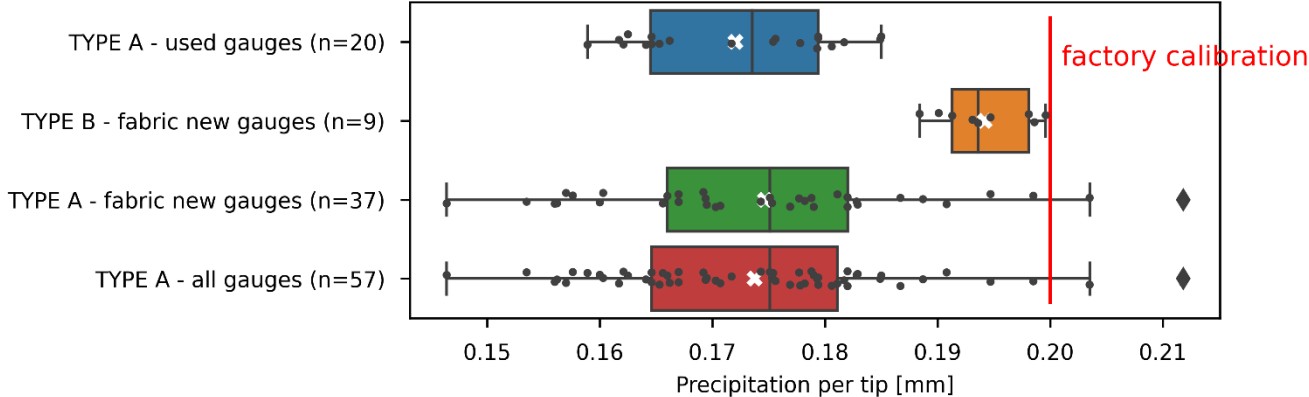

**Figure 3: Comparison between used and fabric new gauges – precipitation needed per tip. Mean for each group depicted with "x". Factory calibration claimed by manufacturer is shown with red line.**

The measured average rain amount to tip the bucket is lower than the stated value of the manufacturer (0.2 mm) for both types,
although the average of Type B is only 2.9% off compared to -13.2% for Type A. The measurements in the laboratory were taken without the rain collector and thus are not accounting for loss-effects of evaporation and wind.

The measurements of both Type A groups (used and new gauges) show large differences between the left and right side of the tipping bucket. This leads to larger errors for small precipitation events (Figure 4). Nevertheless, the influence of the different sensitivities of the respective left and right sides decreases continuously with an increasing number of tips during a precipitation
event and eventually approaches the mean value of all left and right tips of the gauge. To minimize this influence Marsalek

(1981) states that a standard deviation of less than 2% of the mean is desirable – no gauge of Type A could satisfy this need.

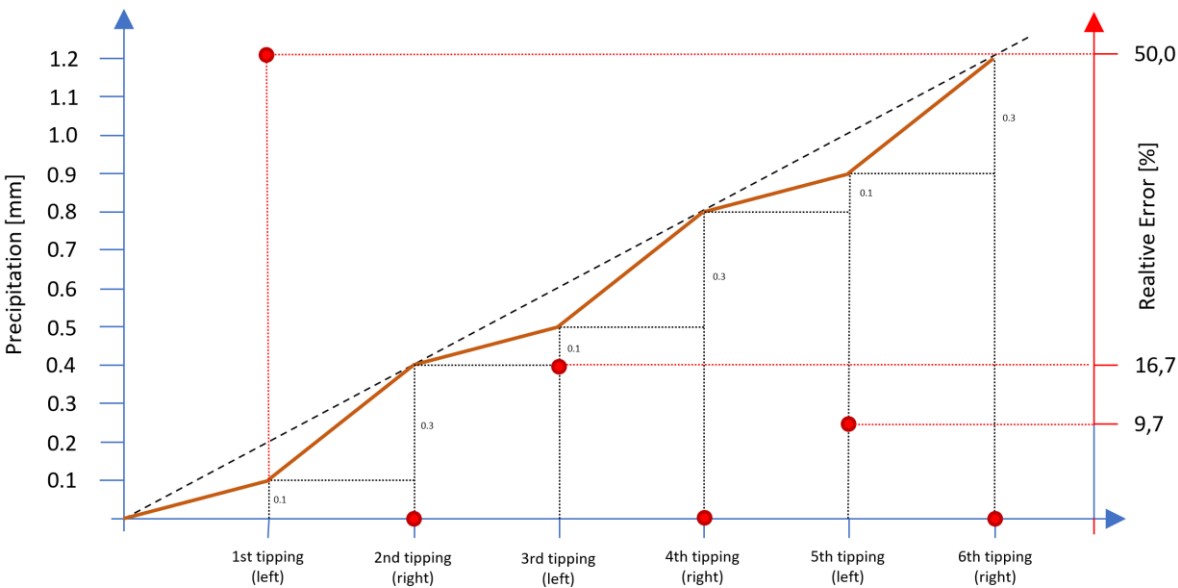

**Figure 4: Development of relative errors due to difference between left and right tippings – relative error is decreasing with each consecutive tipping within a rain event (error: red dots, from left to right; cumulative precipitation – red line)**

Although no bias due to tipping bucket sides is an issue for Type B, differences between individual measurements still exist. However, this variation is much smaller than the differences, which could be observed within the Type A group. While the average SD of the single measurements for Type A gauges was 5.6% (new) and 4.7% (used) of the mean, it was only 1.6% for Type B gauges (see Figure 5). Seven out of nine tested gauges of Type B fulfilled the threshold of 2% stated by Marsalek (1981).

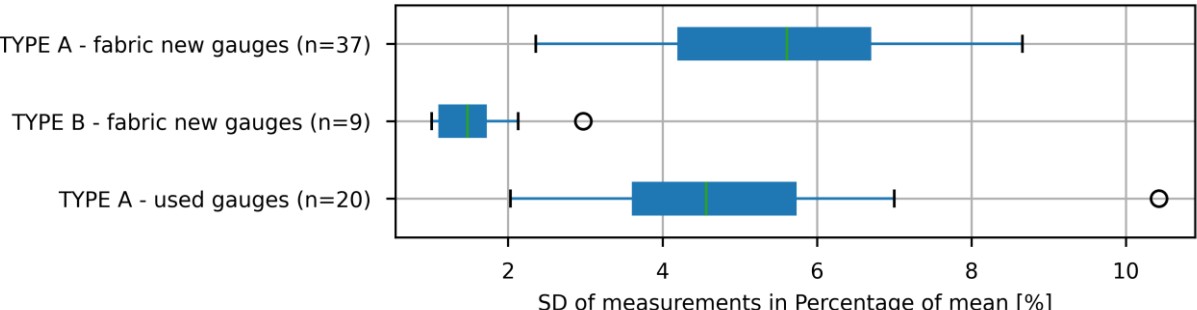


**Figure 5: Comparison between used and fabric new gauges – SD of single measurements in percentage of mean**

## 3.3    Field study

Precipitation data was collected for the period from 01 August 2018 to 30 April 2019. As there were some gaps in the data for
the professional gauges in August 2018 and at the end of April 2019, the data for the whole dataset was analysed for the period
from 01 September 2018 to 25 April 2019. Additionally, there were rare occurrences of gaps in the time series of the *Young*
tipping gauge (28.10.18, 21.02.19, 28.02.19), which could not be explained. Those days have been removed from the dataset
for all gauges.

Figure 6 shows the plot of cumulated precipitation for the whole investigation period. The type A rain gauges show on average
less underestimation (385.85 mm, -11.1%, SD = 17,0 mm) than the other automatic rain gauges when compared to the
*Hellmann* gauge (433.9 mm). While the results of the *Ott Pluvio* are slightly worse than the results of the low-cost gauges
(353.9 mm, -18.4%) the results of the *Young* tipping gauge are considerably worse than the low-cost gauges (310.8 mm, -
28.4%).

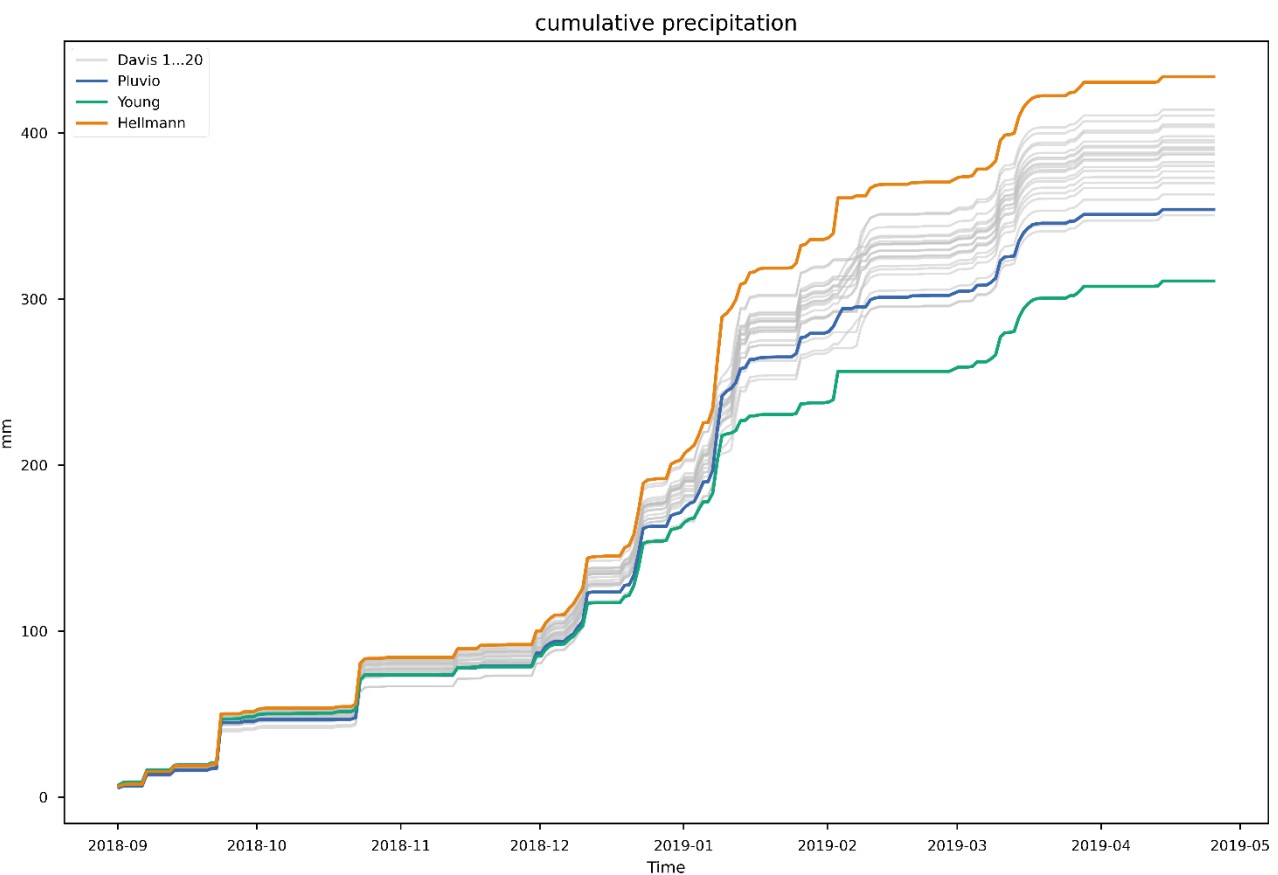

**Figure 6: Cumulative precipitation for all low-cost gauges and three professional gauges for whole study time**

The measured sums of the 20 low-cost gauges are ranging from 350.4 mm, which is slightly less than the *Ott Pluvio*, to 414.0 mm, which is slightly less than the sum of collected precipitation of the *Hellmann* gauge. The distribution of the measurements of the low-cost gauges compared to the professional gauges is shown in Figure 7.

We used the obtained daily error standard deviations (see 2.3.2) to run a Monte Carlo simulation (n = 100) with the daily time series for each professional gauge to obtain a synthetic distribution of cumulated values for each gauge. These distributions were compared with the distribution of the 20 low-cost gauges utilizing a t-test. This resulted in a rejection of the null hypothesis for all professional gauges. Thus, all reference gauges are significantly outside of the distribution of low-cost gauges.

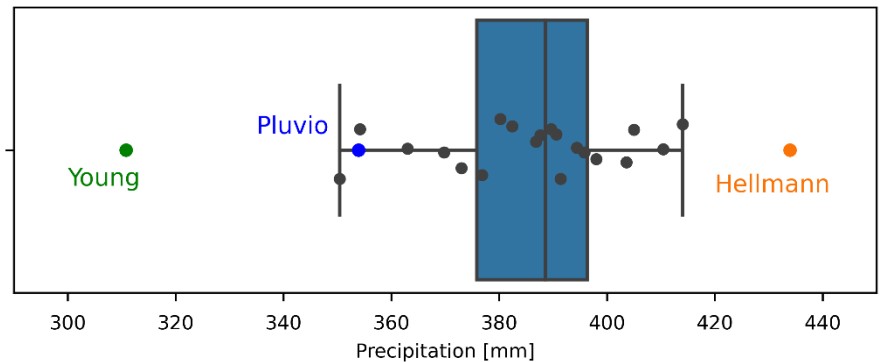

**Figure 7: Distribution of cumulative precipitation of 20 low-cost and 3 professional gauges**

A part of the variation in the results of the low-cost gauges can be explained by the lack of recalibration prior to their usage. In the lab-analysis, calibration values between 0.159 mm and 0.185 mm were obtained, which corresponds of a range between -7.6 % and +7.6 % of the mean of all gauges. This is in the same range as the measured precipitation (-9.3 % to +7.6 % of the mean of all gauges). Nevertheless, calibration values and observed precipitation totals show only a moderate negative correlation Figure 8. For example, gauge 14 recorded the least number of tips and its correlation value (amount of water needed per tip) was the highest. In contrast gauge 10 and gauge 13 measured about the same amount of precipitation (tippings) – but their calibration values are in completely different ranges.

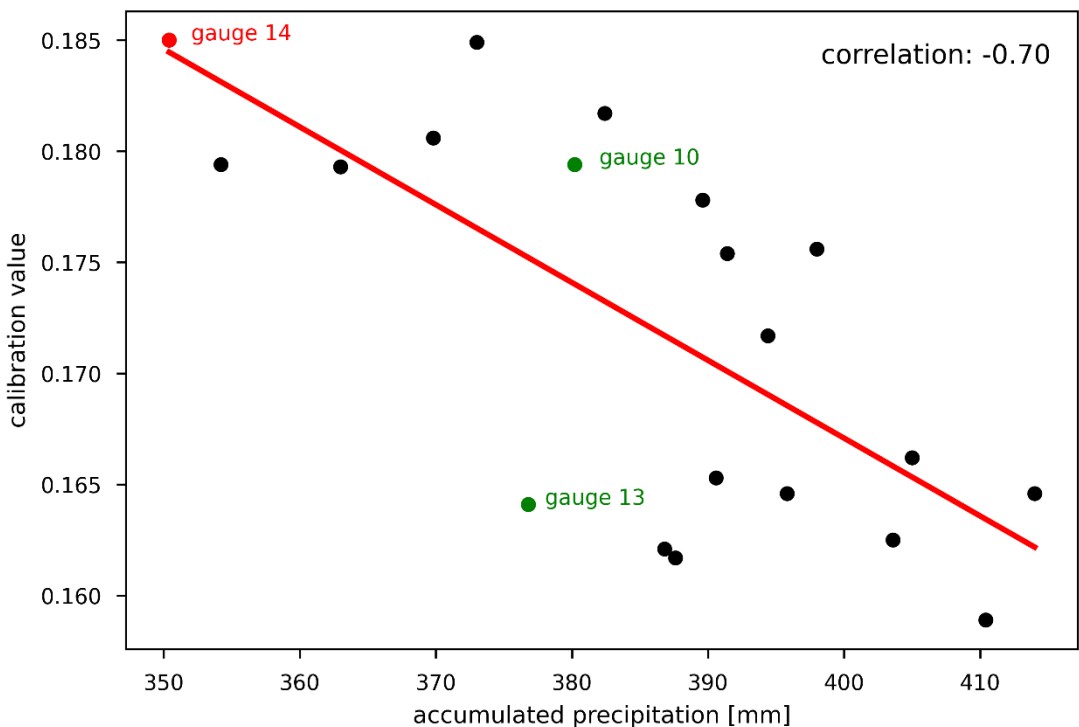

**Figure 8: Calibration values vs measured precipitation of the 20 gauges used in the field study (TYPE A)**

Multiplying the calibration values of each gauge with the tipping's recorded by the associated gauge would result in a much lower measured precipitation (due to the lower mean of 0.172 mm). However, the standard deviation would also be much smaller (12.5 mm vs 17.5 mm, Figure 9). Thus, to improve the issue of the scattered calibration values, one could then scale the mean of the calibration values to the claimed value by the manufacturer of 0.2 mm and subsequently all values by this factor. This would then result in a higher precision of the obtained precipitation for all gauges. Nevertheless, this step is only

possible for a larger group of gauges at hand.

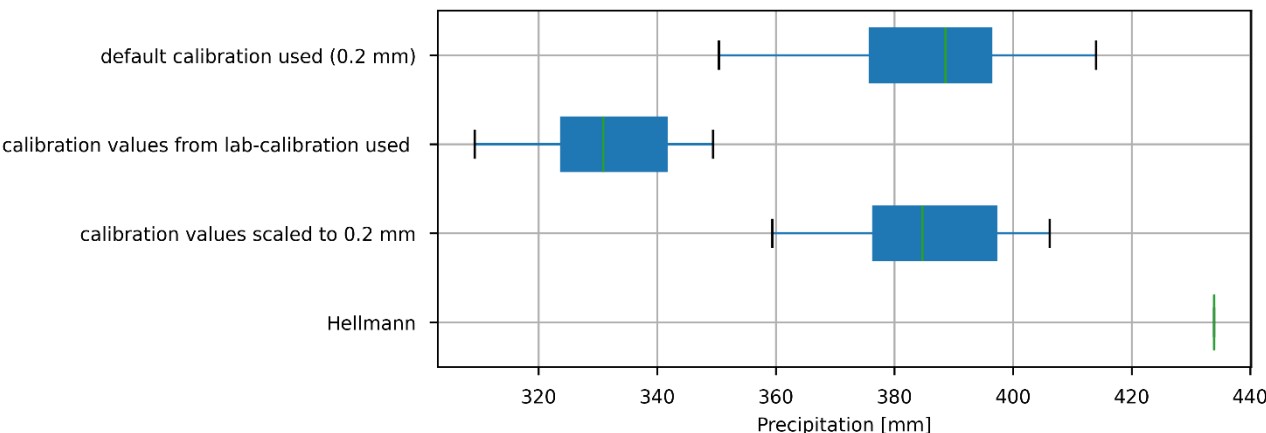

**Figure 9: Comparison between different calibration values used.**

Comparing the cumulated rainfall values of the different gauges in more detail, two periods of strong deviations of the low-cost gauges to the professional gauges can be seen. This observation can be made for all low-cost gauges and is caused by partial blocking of the funnel with snow on 08 January 2019 and 03 February 2019 (see Figure 10 – showing cumulative rainfall with hourly resolution). On 08 January 2019, rising temperatures lead to simultaneous snow melting in all gauges and a subsequent run off and registration of the melting snow on 12 January 2019 and the following days. On 03 February 2019 a warming of the black funnel due to solar irradiation led to a slightly different pattern of precipitation measurement. This time, the gauges at the edge of the array were emptied first (#1-5, 10, 15, 20), while the ones inside the array were blocked longer (e.g., #6-9, 11-14, 16-19) until 08 February 2019.

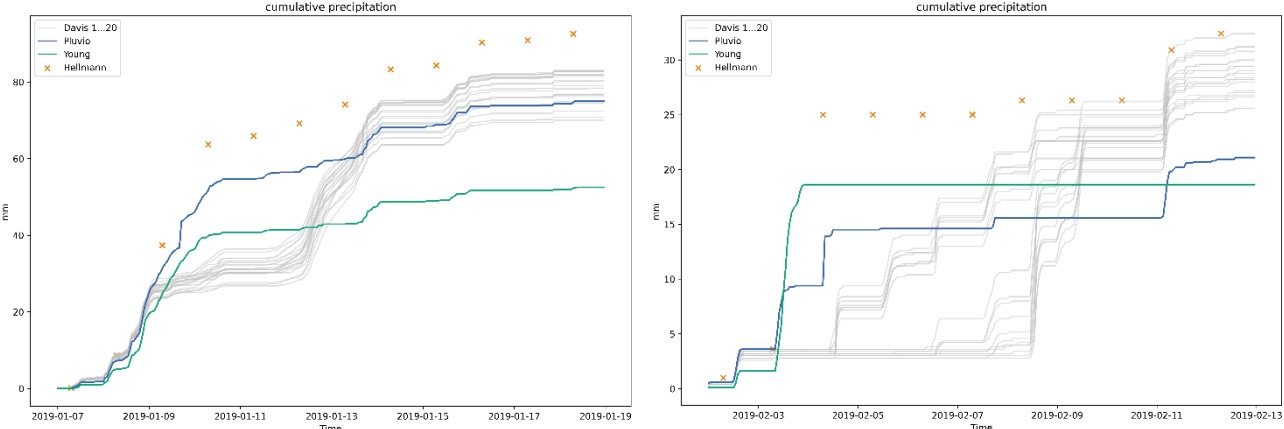

**Figure 10: Cumulative rainfall for two selected periods with hourly resolution. Note, hourly data for the Hellmann gauge is not available and hence is only marked with the daily readings (orange crosses).**

Although results of cumulated precipitation over the whole testing period are promising, Pearson correlation coefficient values for shorter timeframes are not. The correlations for daily values between the different low-cost gauges are high (0.937…0.997).

However, the correlations with the professional gauges are significantly lower (0.779 - 0.841 for Hellmann, 0.730 - 0.784 for *Young* and 0.828 - 0.890 for the *Ott Pluvio*). The low correlation values can partly be explained by the circumstance that the

low-cost gauges have no heating for handling snow precipitation. Removing the days with blockage and subsequent melting of water from the dataset (09 January – 15 January / 03 February – 10 February) yields a strong increase of the correlation values (daily) with the professional gauges (0.973 – 0.990 for Hellmann, 0.897 – 0.912 for *Young* and 0.973 – 0.991 for the *Ott Pluvio*). Nevertheless, correlation values within the group of low-cost gauges are higher compared to the professional gauges, ranging from 0.977 to 0.999 for daily values.

Measuring principles (weighing vs. tipping gauge) and resolutions (0.1 mm vs. 0.2 mm) differ between *Ott Pluvio* and the low-cost gauges. Therefore, correlation values were calculated for a wide range of accumulation intervals (one minute to one day) to assess which accumulation intervals might be needed for acquiring reliable data. As expected, correlations increase with an increased accumulation time. As already shown for the daily values, correlations within the group of low-cost gauges are higher than correlations between low-cost gauges and *Ott Pluvio*. Nevertheless, differences decrease with increasing

accumulation intervals (Figure 11).

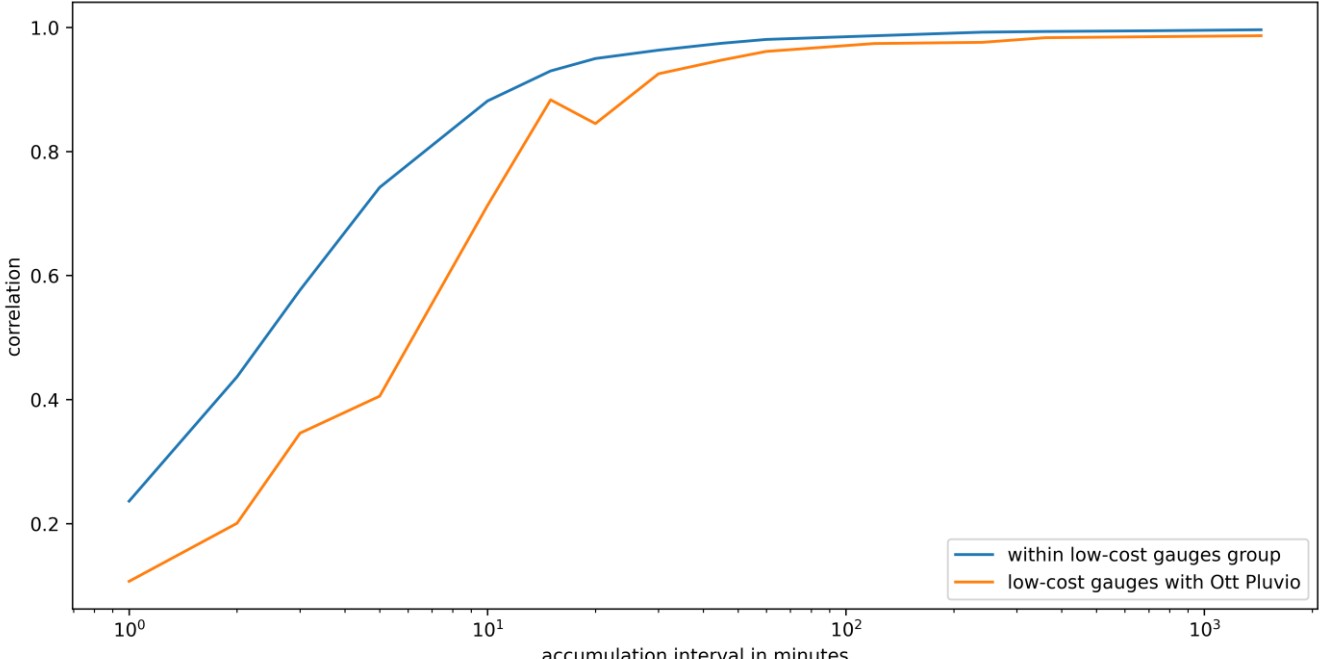

**Figure 11: Average Correlation values for different accumulation intervals (low-cost vs low-cost and low-cost vs Ott Pluvio)**

As the correlation of the low-cost gauge with the *Ott Pluvio* is very low for accumulation intervals of 15 min and less, one could try to use several low-cost rain gauges to overcome this problem. We investigated this by randomly choosing a subset

of gauges (1 to 5) from the dataset and subsequently calculating the correlation of their average measurement with the reference gauge for 2000 simulations. Results for 1, 5 and 15 min can be seen in Figure 12. It can be shown that an increase of used

gauges leads to increased correlations. The biggest gain can be seen when changing from one to two gauges. The effect than diminishes with further gauges. Nevertheless, the benefit of added gauges is a lot smaller compared to increasing the accumulation interval.

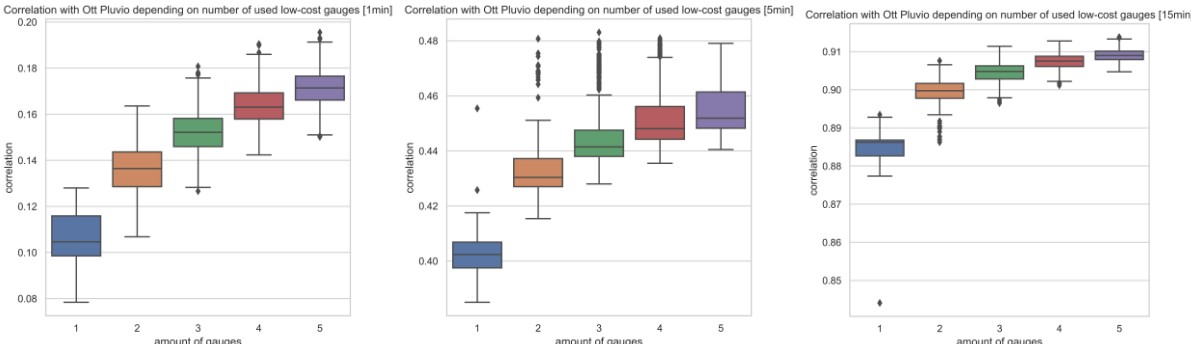


**Figure 12: Correlation for average rainfall measured by subset of 1-5 low-cost gauges with Ott Pluvio for 3 different accumulation intervals (from left to right: 1 minute, 5 minutes and 15 minutes)**

## 4    Discussion

In this study, we assessed the suitability of a widely used low-cost precipitation sensor for an out-of-the-box use, i.e., applying
the sensor in the field without prior calibration. The results of the lab calibration suggest that it is beneficial to check the factory calibration for the older type of the sensor (Type A) for outliers.

### 4.1    Lab-Calibration

It could be shown for Type A gauges that the mean of two consecutive tipping's is consistent across all tested gauges resulting in a mean of 0.174 mm per tip. However, it also became obvious that some tipping gauges deviated strongly from this mean
value. Utilizing the presented analysis scheme outliers from the expected mean can be detected after approx. 4 to 5 tilts per side. Furthermore, the amount of water required for a tipping of one of the buckets is important. In the case of large differences between both bucket sides, the measurement accuracy of very small precipitation events, in which only a few tilts occur, will be affected more strongly, resulting in larger relative errors. However, this is only the case for odd numbers of tipping. A re-calibration of the gauge is recommended if the difference between both bucket sides is larger than 0.05 mm of precipitation
(25 % of nominal volume). In contrast, Marsalek (1981) advised to continue the calibration until the mean was within of 2 % of the nominal volume.

The measured means of the tipping are consistently lower than 0.2 mm across all gauges (Type A and B) leading to the assumption that a factory calibration might have been performed. Furthermore, potentially only one side of the mechanism (Type A only) had been calibrated until the desired volume was met, which could have led to the observed differences between
the two buckets. Due to the design of a tipping bucket, water is lost when the bucket is tilted, as the other bucket is not in position again fast enough. This can lead to an under-catch, which increases with the rainfall intensity. These errors can range

from 10 % to 30 % (Humphrey et al., 1997; Marsalek, 1981). Furthermore, error influences due to evaporation and wind are conceivable.

In contrast to the results of gauges of type A, the measured mean for type B (0.194 mm) is considerably closer to the nominal
volume (0.2 mm). Also, the deviations between the single measurements are much smaller compared to type A. These gauges might have been factory calibrated to a higher volume because the mechanism is less prone to under-catch at higher rainfall intensities.

## 4.2    Field-study

In the field study it could be shown, that the gauges of type A in average show accumulated rainfall results, which are closer
to the Hellman reference gauge, than the two other professional gauges. Nevertheless, the spread of the measured precipitation totals is about twice as big as stated in the manufacturer's datasheet (± 4 %), with deviations ranging from - 9.3 % to + 7.6 % compared to the mean of all gauges. Even the SD for alle gauges (4.6 %) is bigger than that value. The manufacturer is stating, that this accuracy is valid for rain rates up to 100 mm hr$^{-1}$, which have not been observed by any of the gauges in the study period.

Parts of this variability can be explained by the lack of calibration before use as shown above.

Further, other error sources can have an impact on the measured precipitation. As the 20 gauges have been placed in a very dense array, some of the gauges are more shielded from the wind compared to others. This should lead to an under-catch for the exposed gauges, when compared to the more shielded ones (Pollock et al., 2018). Nevertheless, this systematic behaviour was not observed.

Splashing of raindrops and thus rainfall being diverted to adjacent gauges might lead to a systematic pattern in the dense array. However, this was not observed in this study.

The dense array could have an influence on evaporation behaviour, as some gauges are more exposed to direct sunlight, than others. This is especially true in the winter months, as the sun only rises to very low angles, which was already seen in the behaviour of the melting snow (as described above).

A further source of error might also be the non-optimal mount of the gauges. To ensure good readings, the gauges have to be perfectly levelled (Burt, 2009). While installing the array, the levelling was done using the provided circular bubble level, which has a limited accuracy. Furthermore, the frame on which the gauges were installed was made of wood, which might degrade and thus move over time.

Last but not least, although the array only covered an area of about 1 square meter, small scale variability of rain fields could
also play a role for obtaining different precipitation readings. Nevertheless, other studies, e.g., the WMO Field Intercomparison of Rain Intensity Gauges (Lanza and Vuerich, 2009) used similar or bigger sized setups than this study.

# 5    Conclusions

In this study two low-cost sensor systems based on the Arduino and Raspberry Pi families have been presented. Here it could be shown that utilizing widely available open hardware allows the user to flexibly create a sensor system tailored to the needs on a given site, while keeping the costs low. The presented systems are capable to log sensor data of various sensors and are able to either log data locally or transmit the data to the internet.

Further the factory calibration of the widely used *Davis* tipping rain gauge was examined, both in the laboratory and in the field. In the lab, different generations of the gauge have been tested. Here, the results show a distinct difference between the old and the newer generation of the gauge. While for the older type in average 0.174 mm of water was needed for a tip, 0.194 mm was needed for the newer generation - which is less than officially stated by the manufacturer. For the older type it was also found, that the gauges are not tipping equally with both buckets. This leads to errors in the precipitation measurements for small rain events. To minimize this error, gauges should be checked and calibrated for equal tipping before installation. This is particularly important for the probably already several ten thousand gauges of TYPE A in use worldwide. Here the factory calibration should at least be checked. For the newer TYPE B it could be found, that the deviation between the tippings is much smaller compared to the old type. If this has a positive impact on the accuracy was not within the scope of this study, but has to be further investigated. Nevertheless, for these gauges, the factory calibration should also be checked before use, as our sample was very small (n=9).

Beside the lab calibration, 20 gauges (TYPE A) have been tested against professional rain gauges in the field. Here, the results showed an undercatch when compared to the reference Hellmann gauge of the climate station where the array was installed. Further, it could be shown that the spread of the factory calibration is larger (by magnitude of 2) than stated by the manufacturer. Nevertheless, results of the low-cost gauge have been closer to the reference than *Young* Tipping gauge and *Ott Pluvio* respectively. Our results suggest (high correlation for longer accumulation intervals vs. professional gauges), that the error due to undercatch could be mitigated by applying a factor for each gauge – however, this would require measurements against a reference station, either in the field or in the lab.

The study has shown that the tested low-cost sensor is suitable for use in the collection of meteorological data. However, the factory calibration should at least be checked, if not recalibrated before use. Paired with a low-cost sensor system and properly set up, these sensors can be beneficial for the densification of existing sensor networks. For accurate results, the desired out-of-the-box use is not recommended.

**Data availability**

Datasets obtained during the experiments (Lab and Field) as well as source code are available through Zenodo: https://doi.org/10.5281/zenodo.10838614 (Krüger, 2024). Source Code will be updated and is available through github[2].

---

[2] https://github.com/kruegertud/tharandt_raingauge/

**Author contribution**

RK and PK developed the concept and methodology for the study. RK was programming the code for the sensor systems and carried out the investigation und subsequent formal analysis. PK supported in setting up and carrying out the experiments. RK was creating the visualizations and writing the original draft of this paper. PK and AE were reviewing the paper and gave continuous support during the study.

**Competing interests**

AE is a member of the editorial board of journal Geoscientific Instrumentation, Methods and Data Systems. The peer-review process was guided by an independent editor, and the authors have also no other competing interests to declare.

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
