# Peer review of "Calibrating low-cost rain gauge sensors for their applications in IoT infrastructures to densify environmental monitoring networks"

_Geoscientific Instrumentation, Methods and Data Systems, 2023_

## Author Response (AR1)

Legend:

Referee comments in grey, **Authors response in Black**, *Authors changes with line numbers in blue.*

Review of "Calibrating low-cost rain gauge sensors for their applications in IoT infrastructure to densify environmental monitoring network" by Krüger et al

Review by Rolf Hut

The authors calibrated a collection of off the shelf low-cost rain gauges to test if they are usable in scientific applications with the factory calibration. Given the amount of projects that aim to use Personal Weather Stations (PWS) to supplement professional networks, this is a valuable addition to the literature. I do have, however, some suggestions to in my opinion improve the paper (and its usability by the scientific community) before publication.

**Thanks for time and effort to review our manuscript. We have replied inline in the text. Author comments are in black, reviewer comments are grayed out.**

Overall comments

The manuscript as written hinges on two thoughts: on the one side a lab and field calibration of low-cost rain gauges and on the other side an overview of IoT hardware and cost needed to use low cost sensors in general and rain gauges in particular. This last part (IoT hardware) is worked out in far less detail compared to the first part (rain gauge calibration). In literature a large collection of articles reviewing state of the art development boards (including Arduino and Raspberry Pi) for use in environmental sensing in general and weather stations, is available. I would suggest that the authors focus on the rain gauge calibration and remove, or move ta an appendix, paragraphs 2.1 until and including 2.2.4. In the main text the authors can cite relevant literature on IoT hardware reviews. (a quick search on google scholar already resulted in these DOIs, there is much more: https://doi.org/10.3390/ijerph17113995, https://doi.org/10.1016/j.cosrev.2021.100364, https://doi.org/10.1016/j.procs.2014.07.059, 9734/AJRCOS/2021/v9i130215)

**We agree, that the paragraph on the used IoT hardware is relatively short. Nevertheless, although many studies have already been published on using IoT capable developer boards, we wanted to include the used setup. Here the aim is to increase applicability of the rain gauges related findings and a better transferability for the final user. The source code will be added to appendix.**

*Line 140ff (p6):     Further, a Raspberry Pi based system has 140 been used in conjunction with a SHT31 temperature/humidity sensor, a pyranometer and a*

*Davis Vantage Pro2 rain gauge. The source code for those setups is available through Zenodo (Krüger, 2024).*

In calibrations of rain gauges the crucial question is always: "what do we use as 'the truth' and the authors have three reference devices available. They choose to use the Hellman gauge as reference without further justification. I would ask the authors to substantiate why the Hellman, compared to the other devices, should be considered "reference" (or "thruth").

**Thank you for the remark – indeed the decision here is not easy. First of all, the Hellmann device was used as reference because it's considered as the reference for the climatological measurements at this meteorological site since the 1950s. This fact is already stated in the manuscript, but not yet made clear as justification. Further, the used instrument principle requires no mechanical and electronic parts and thus the data quality should be stable, as the instrument is set up properly. For comparisons on timeframes shorter than 1 day, the Ott Pluvio will be used (see below). Will be clarified in the text.**

*Line 213ff (p8): The Hellmann device is used as the reference for the climatological measurements taken at this station since 1951 (Fig 1) and thus considered as the reference instrument in this study. The Hellmann gauge requires no mechanical and electronic parts, thus data quality should be stable, as the instrument has been set up properly.*

Continuing on point 2: the authors only report on the difference between the different gauges, both within the groups of low-cost gauges and between the low cost gauges and the reference gauges. However, they do not quantify if these are significant in the light of uncertainties, either inherent in their way of measuring, or inherent in the nature of rainfall. Given the large amount of low-cost rain gauges they use, it should be straightforward to indicate if the values of the the reference gauges are significantly outside the distribution of low-cost gauges. If the authors have access to a long time series from the reference devices (which I assume they have), they could use triple co-location to estimate the uncertainties in the three reference devices. This would than allow for a two-way comparison between the reference devices and the low-cost gauges. There is a wealth of literature on (how to do) comparisons between rain gauges, including the statistics involved. I suggest starting at Lanza 2009 (https://doi.org/10.1016/j.atmosres.2009.06.012)

**Thanks for the suggestion of the triple collocation method. We used a longer time series of the three reference gauges ranging from 2017 to 2019 consisting of daily observations to estimate the uncertainties. Inspection of the three scatterplots with all combinations of the reference gauges led to the assumption that the Ott Pluvio is the best performing as the Hellmann/Young Scatterplot had the lowest correlation (Stoffelen and Vogelzang, 2012). We**

then used an implementation provided by Jur Vogelzang[1] to estimate the error variances using the Ott Pluvio as reference system.

The following daily error standard deviations could be determined: $std_{Pluvio}$= 0.150mm/d, $std_{Hellmann}$ = 0.183mm/d, $std_{Young}$ = 0.278mm/d

[Figure]

*Line 224ff (p9): Even with three professional gauges, the true precipitation remains unknown as each of these devices area also associated with uncertainties. Triple collocation analysis (Stoffelen and Vogelzang, 2012; Stoffelen, 1998) can be used to estimate the error 225 variances of these three independent, but collocated datasets without requiring knowledge of the true precipitation amount. We used a longer time series of the three professional gauges lasting three years (from 01.01.2017 to 31.12.2019), consisting of daily observations, to estimate the uncertainties. Inspection of the three scatterplots (Figure 1) with all combinations of reference gauges led to the assumption that the Ott Pluvio is the best performing, because the scatterplots for the Hellmann and Young devices revealed lower correlations (Stoffelen and Vogelzang, 2012). Therefore, we used an implementation 230 provided by Jur Vogelzang[1] to estimate the error variances with the Ott Pluvio as reference system. Subsequently, the daily error standard deviations could be determined: $stdPluvio$ = 0.150mm $d^{-1}$, $stdHellmann$ = 0.183mm $d^{-1}$, $stdYoung$ = 0.278mm $d^{-1}$. These values were used to evaluate the results of the low-cost gauges when compared with the reference gauges.*

We then used a Monte Carlo simulation utilizing the daily datasets of the field campaign and the daily uncertainties of the rain gauges to generate a distribution of artificial datasets for each gauge. These have then been compared with the distribution of low-cost gauges utilizing a t-test, resulting in a rejection of the null hypothesis for all reference gauges. Thus, all references gauges are significantly outside of the distribution of low-cost gauges. These steps will be added to the manuscript.

*Line 317ff (p14): We used the obtained daily error standard deviations (see 2.3.2) to run a Monte Carlo simulation (n = 100) with the daily time series for each professional gauge to obtain a synthetic distribution of cumulated values for each*
* * *
[1] https://github.com/knmiscat/triple_collocation

*gauge. These distributions were compared with the distribution of the 20 low-cost gauges utilizing a t-test. This resulted in a rejection of the null hypothesis for all professional gauges. Thus, all reference gauges are significantly outside of the distribution of low-cost gauges.*

In the lab calibration it is extremely important that the rain gauges are placed perfectly horizontal. I assume the authors made sure of this. I would suggest to add a few sentences on how this was done. Furthermore, it is important to know if all rain gauges were oriented exactly the same direction on the table. If the table was even slightly tilting, having all rain gauges in the same orientation would result in a bias towards a certain direction and could explain the left-right difference observed?

**In preparation of the lab calibration the table was levelled utilizing the adjusting screws in the table legs. The rain gauges themselves have been levelled using the built-in bubble level. Further, slices of paper have been used to account for remaining unevenness on the table.**

**During the calibration, all gauges have been oriented in the exact same direction**

**More detailed explanation will be added to the manuscript as suggested.**

*Line 177ff (p7): In preparation of the lab-calibration the table was levelled utilizing adjusting screws at the table legs. The rain gauges themselves have been levelled using a built-in bubble level. Further, slices of paper have been used to account for remaining unevenness on the table. During the calibration, all gauges have been oriented in the exact same direction on the table.*

The analyses done within the discussion is, in my point, central to the manuscript. I would suggest to move the results of the comparison of the field and lab experiment to the result, explain in the methods which (statistical) methods are used to compare the two datasets and in the discussion only reflect on the result, not present new ones.

**Thank you for critically pointing that out. Will be restructured.**

*Parts of this analysis have been moved to line 324ff (p14) (in 3.3 results / Field study)*

Specific comments

All figures need more detail in their captions to understand what is shown.

**Will be improved.**

**Several additions to the figure captions.**

In figure 1 I would add a vertical (red?) line at 0.20 mm to indicate where the factory calibration is.

**Will be added.**

*Has been added (to now figure 3 (p11))*

Figure 6 could use the Hellman data as crosses or points. Especially on the one hourly data it is interesting to look at the uncertainties of the three reference devices (see above).

**Will be added.**

*Crosses for the Hellmann cumulated daily values (at 7 o'clock MEZ) have been added to both figures (p16).*

Overall I think this is a highly relevant paper given the focus on citizen science projects to use Personal Weather Stations to supplement professional networks. With the above suggestions implemented I would be happy to recommend publication in GI.

Rolf Hut

References:

Stoffelen, A. and Vogelzang, J.: Triple collocation, https://doi.org/10.13140/RG.2.2.30926.66888, 2012.

Review of "Calibrating low-cost rain gauge sensors for their applications in IoT infrastructure to densify environmental monitoring network" by Krüger et al

The paper addresses the calibration of low-cost rain gauges in laboratory and field evaluation with referenced ones.

It is of interest for field application to enrich dataset and reducing instrumentation cost.

**Thanks for time and effort to review our manuscript. We have replied inline in the text. Author comments are in black, reviewer comments are grayed out.**

The part concerning low-cost data acquisition system could be shorten, as different publications are available in literature. Anyway, a paragraph focus on time synchronization versus the reference system used in the field test would be an interesting complement.

**Time synchronisation of the low cost sensor system with the reference gauges was ensured as time for both type of systems was set through network time protocol (NTP). Will be clarified.**

*Line 247ff (p10): Time synchronisation of the low-cost sensor system with the reference gauges was ensured by setting the time for both type of systems through the network time protocol (NTP).*

**We like to have the low-cost acquisition part in the manuscript to increase applicability of the rain gauges related findings and a better transferability for the final user. The source code will be added to appendix.**

*Line 140ff (p6):      Further, a Raspberry Pi based system has 140 been used in conjunction with a SHT31 temperature/humidity sensor, a pyranometer and a Davis Vantage Pro2 rain gauge. The source code for those setups is available through Zenodo (Krüger, 2024).*

Concerning the laboratory calibration, the lecturer would appreciate to have a plot of the distribution of results obtained for all your rain-gauges, supposed to be Gaussian. Is the median close to your mean value?

**There is a plot on page 10 (fig. 1) showing the distribution of the two different types. A Boxplot for all Type A gauges combined will be added, further the single measurements will be visualized like in figure 5 (p12).**

*Line 267ff (p11): Figure 3 has been altered as proposed. Mean values are now also shown in the figure. Further, the value for the factory calibration is shown in the figure.*

**Median Values will be added to the text. Median of all gauges (type A) is 0.1751mm (mean = 0.1737mm).**
**Distribution of results is indeed Gaussian for all 4 groups (A – new, A – used, A – all, B) – tested with Kolmogorov–Smirnov test (alpha = 0.05)**

*Line 263ff (p11): The distribution of results in all four groups (A – new, A – used, A – all, B) was tested if it is Gaussian, using the Kolmogorov–Smirnov test (alpha = 0.05). This could be confirmed for all four groups.*

During laboratory experiments, the flatness and horizontality was supposed to be controlled, what about the field conditions.

**Horizontality of the rain gauges was ensured by the usage of the built-in bubble level of the rain gauges and the usage of washers while fixing the gauges to the (levelled) frame.**
**I will add the explanation to the manuscript.**

*Line 239ff (p9): The frame is levelled and set at a height of about 1 m above ground to match to the height of the reference gauges, located about 10 m to the south of the reference 240 instruments. Horizontality of the rain gauges was ensured by the usage of the built-in bubble level of the rain gauges and the washers while fixing the gauges to the (levelled) frame. The setup is shown in Figure 2.*

What was the confidence interval during your laboratory calibration?

**Confidence intervals for a confidence level of 95% have been calculated as follows (A-new: 0.1680mm...0.1795mm; A – used: 0.1699mm...0.1795mm; A – all: 0.1704mm...0.1771mm; B: 0.1911mm....0.1972mm). Will be added to the text.**

*Line 266ff (p11): Confidence intervals for a level of 95% have been calculated as follows: A-new: 0.1680 mm - 0.1795 mm; A-used: 0.1699 mm - 0.1795 mm; A-all: 0.1704 mm - 0.1771 mm; B: 0.1911 mm - 0.1972 mm).*

The referenced station for field trials are not positioned at the same spatial location, how do you controlled the spatial homogeneity? It could add uncertainty to your field results to be taken into account for the analysis.

**We agree, that small scale variability of precipitation could add uncertainty to the results. Nevertheless, we didn't control or account for spatial homogeneity as the gauges have been set up close to the reference (10m), well within the measurement site. Further, other gauges are also distributed to the measurement site.**

**Other studies, f.e. the WMO Field Intercomparison of Rain Intensity Gauges (Lanza and Vuerich, 2009) used similar or bigger sized setups without accounting for spatial homogeneity.**

*Line 441ff (p19): Last but not least, although the array only covered an area of about 1 square meter, small scale variability of rain fields could also play a role for obtaining different precipitation readings. Nevertheless, other studies, e.g., the WMO Field Intercomparison of Rain Intensity Gauges (Lanza and Vuerich, 2009) used similar or bigger sized setups than this study.*

As the final aim is to enhance the amount of sensor using low-cost sensor, using opportunistic data from private owner of rain stations can be discussed by comparison with the knowledge acquired during your experiments.

**It is true that this sensor type is widely used by local authorities and private users. Although precipitation data from that kind of sensors can be acquired through private weather networks like wunderground or else, analysing those datasets is difficult without further informations on the specific set up, and thus beyond the scope of this study.**

The research work presented is of importance for field experiments.

I hope that these few suggestions will help authors to improve their paper for publication in GI Journal.

References

Lanza, L. G. and Vuerich, E.: The WMO Field Intercomparison of Rain Intensity Gauges, Atmospheric Research, 94, 534–543, https://doi.org/10.1016/j.atmosres.2009.06.012, 2009.